# Placental Growth Factor and Pregnancy-Associated Plasma Protein-A as Potential Early Predictors of Gestational Diabetes Mellitus

**DOI:** 10.3390/medicina59020398

**Published:** 2023-02-17

**Authors:** Vesselina Yanachkova, Radiana Staynova, Teodora Stankova, Zdravko Kamenov

**Affiliations:** 1Department of Endocrinology, Specialized Hospital for Active Treatment of Obstetrics and Gynaecology “Dr Shterev”, 1330 Sofia, Bulgaria; 2Department of Pharmaceutical Sciences, Faculty of Pharmacy, Medical University of Plovdiv, 4002 Plovdiv, Bulgaria; 3Department of Medical Biochemistry, Faculty of Pharmacy, Medical University of Plovdiv, 4002 Plovdiv, Bulgaria; 4Department of Internal Medicine, Medical University of Sofia, 1431 Sofia, Bulgaria; 5Clinic of Endocrinology, University Hospital “Alexandrovska”, 1431 Sofia, Bulgaria

**Keywords:** gestational diabetes, placenta, pregnancy associated plasma protein-A, placental growth factor, biomarkers, insulin resistance

## Abstract

Gestational diabetes mellitus (GDM) is one of the most common pregnancy complications and one of the main causes of adverse pregnancy outcomes. An early diagnosis of GDM is of fundamental importance in clinical practice. However, the major professional organizations recommend universal screening for GDM, using a 75 g oral glucose tolerance test at 24–28 weeks of gestation. A selective screening at an early stage of pregnancy is recommended only if there are maternal risk factors for diabetes. As a result, the GDM diagnosis is often delayed and established after the appearance of complications. The manifestation of GDM is directly related to insulin resistance, which is closely associated with endothelial dysfunction. The placenta, the placental peptides and hormones play a pivotal role in the manifestation and progression of insulin resistance during pregnancy. Recently, the placental growth factor (PlGF) and plasma-associated protein-A (PAPP-A), have been shown to significantly affect both insulin sensitivity and endothelial function. The principal function of PAPP-A appears to be the cleavage of circulating insulin-like growth factor binding protein-4 while PlGF has been shown to play a central role in the development and maturation of the placental vascular system and circulation. On one hand, these factors are widely used as early predictors (11–13 weeks of gestation) of complications during pregnancy, such as preeclampsia and fetal aneuploidies, in most countries. On the other hand, there is increasing evidence for their predictive role in the development of carbohydrate disorders, but some studies are rather controversial. Therefore, this review aims to summarize the available literature about the potential of serum levels of PlGF and PAPP-A as early predictors in the diagnosis of GDM.

## 1. Introduction

Gestational diabetes mellitus (GDM) is defined as a disorder of glucose metabolism recognized for the first time during pregnancy, even though the abnormalities may have existed before pregnancy and may persist after the delivery [1]. It has been estimated that approximately 7% of all pregnancies are complicated by GDM, resulting in more than 200,000 worldwide cases annually [2,3]. The prevalence may vary from 1% to 14% of all pregnancies depending on the population studied and the diagnostic tests used [2]. GDM is the most common cause of hyperglycemia during pregnancy, accounting for more than 85% of all cases [3,4]. The remaining 15% are due to pre-existing type 1 or type 2 diabetes mellitus. In addition, GDM is highly associated with many other serious maternal and fetal complications [4,5]. Therefore, more research and clinical efforts should be focused on preventing GDM risk factors and especially on its early diagnosis. The universal screening for GDM includes a 75 g oral glucose tolerance test at 24–28 weeks of gestation, but often then is too late and both GDM and its complications may have already occurred [3,4,5,6,7,8]. On the other hand, selective screening in the early stages of pregnancy is recommended mainly in the presence of risk factors in mothers (maternal overweight or obesity, age, ethnicity and/or family history of diabetes, previous macrosomia, GDM in previous pregnancies) [1,6,9,10,11]. Hence, an important question arises: Whether biochemical markers that are widely tested during the first trimester of pregnancy in terms of screening for aneuploidies, such as placental growth factor (PlGF) and plasma-associated protein-A (PAPP-A), could also be included in the recommendations for GDM screening?

In this narrative review we discuss the potential role of serum biomarkers—PlGF and PAPP-A—as early predictors of GDM diagnosis.

## 2. Pathogenesis of GDM

Pregnancy is a condition of physiological insulin resistance (IR) [12]. The factors contributing to this IR include increased maternal adiposity, decrease in insulin sensitivity, placental hormone production, elevated levels of cortisol and various inflammatory markers such as tumor necrosis factor-α and interleukin 6 [12,13]. IR is a state in which a given concentration of insulin produces a less-than-expected biological effect in target tissues such as adipose tissue, muscle, and liver [12,13,14,15]. As a result, there is reduced glucose uptake, mainly in the muscle cells and adipocytes [14]. The compensatory mechanism is the activation of glycogenolysis and increased hepatic glucose production [15]. In addition, the ability of insulin to suppress whole-body lipolysis is also reduced during pregnancy, contributing to a greater postprandial increase in free fatty acids [16,17,18]. In the initial stages of IR, the reduced tissue sensitivity and response have been compensated by hyperinsulinemia. Gradually, however, the pancreatic β-cells become depleted, their secretory function declines, and hyperglycemia appears [12,15]. The decrease in insulin sensitivity is most pronounced in the second half of pregnancy and its major function is to limit the absorption of glucose by the mother, which meets the needs of the developing fetus. During a normal pregnancy, IR is compensated by the increase of insulin production by the mother’s pancreas [14]. In some pregnant women, especially those with obesity and pre-pregnancy IR, the IR status worsens as the pregnancy progresses, and the pancreatic beta cells cannot continue to maintain the normoglycemia secreting more insulin. Consequently, abnormal blood glucose levels develop, marking the onset of GDM [19].

With the formation of the placenta, several hormones, proteins (cytokines, growth factors, glycoproteins) and other signaling molecules begin to exert their effect on IR [20]. (Figure 1). 

The advance in pregnancy and the growth of the placenta consequently leads to an increased production of these hormones and factors. Since most placental products have insulin-antagonistic effects, IR also worsens [13].

As mentioned above, these changes are usually most noticeable after the second trimester of pregnancy. Therefore, the recommended period for performing GDM screening is 24–28 weeks of gestation [1,7,21,22].

However, during the first trimester (11–14 weeks of gestation) all pregnant women undergo a screening for fetal aneuploidy. This can help to determine whether the fetus is at risk for a chromosomal abnormality and also might be used to assess the risk of preeclampsia. First-trimester screening includes the measurement of several biochemical markers like human chorionic gonadotropin (hCG), PIGF, soluble fms-like tyrosine kinase-1 (sflt-1) and PAPP-A [23]. These markers are already considered as established predictors of chromosomal abnormalities and some maternal complications (e.g., PlGF and PAPP-A are widely used in the screening for pre-eclampsia) [24,25,26]. In addition, first trimester PAPP-A levels have been associated with early prediction of pregnancy-induced hypertension [27]. Furthermore, some recent studies have suggested that PAPP-A and PlGF may also be relevant markers for carbohydrate disorders manifested during pregnancy [24,28,29,30,31].

## 3. PAPP-A

PAPP-A is a zinc-containing metalloproteinase, belonging to the metzincin superfamily, first described in 1974 as a protein present in the plasma of pregnant women [32]. The structure of PAPP-A is similar to that of human placental lactogen and hCG. PAPP-A concentration increases with the progress of pregnancy until delivery [33]. It is produced by the syncytiotrophoblast and acts as a protease for the IGF-binding protein-4 [34]. In addition to the placenta, this protein is also expressed in the fibroblasts, osteoblasts, endothelial cells, or smooth-muscle cells [20]. The primary mechanism of action of PAPP-A is to cleave the molecule of circulating IGFBP-2,4,5 [20,29]. Thus, an improvement in the activity of IGF is achieved. The reduction of PAPP-A concentration is associated with higher levels of IGFBP and low levels of free IGF, respectively [29].

The reason for the reduced synthesis of PAPP-A from the placenta is the disturbance in the trophoblast invasion. Low protease levels during the first trimester in pregnant women without chromosomal abnormalities are associated with adverse perinatal outcomes, including intrauterine growth restriction, miscarriage, and low birth weight [20,35].

The mechanism of the effects of PAPP-A in the pathogenesis of GDM is not fully understood. It has been assumed that women with low PAPP-A levels also have lower IGF levels [35]. IGF, in turn, plays a role in fetal growth regulation. It takes part in the autocrine and paracrine control of trophoblast invasion [36]. IGF is a stimulator of muscle protein synthesis and the utilization of free fatty acids [37]. The action of PAPP-A is considered to be indirect, as it is the reduced level of IGF that leads to hyperinsulinemia, impaired glucose metabolism, and is therefore inversely correlated with the severity of insulin resistance [24,28,29].

Studies have shown that one of the binding IGF proteins in adipocytes, namely IGFBP-5 and the protease PAPP-A, may be involved in the pathogenesis of GDM [29,37] This is due to a change in the regulation of functional levels of IGF, fat stores, and their metabolism.

During a normal pregnancy, the induction of IGFBP-5 in adipose tissue increases the levels of sequestered IGF-1 and IGF-2, and PAPP-A degrades IGFBP-5, resulting in the release of insulin-like growth factors. This leads to angiogenesis and hyperplastic expansion of adipocytes. In women with GDM, insufficient IGFBP-5 levels and possibly decreased PAPP-A levels lead to reduced IGF bioavailability. Inadequate angiogenesis occurs, resulting in adipocytic hypertrophy and decreased capillary density. Low-grade inflammation and lipotoxicity are manifested, respectively, leading to further insulin resistance and impaired carbohydrate tolerance [38] (Figure 2).

In recent years, serious attention has been directed to placental products. Some of them have been considered as prognostic markers for the occurrence of complications during pregnancy. Several studies have shown that low serum levels of PAPP-A in the first trimester of pregnancy could be associated with the onset of GDM and adverse pregnancy outcomes. [28,39,40,41,42,43,44,45,46,47,48,49,50,51,52,53] (Table 1).

However, the cited studies do not use the same criteria for diagnosing GDM. In addition, there is also a difference in age, ethnicity and body mass index of women included in the analysis. Despite this, most of the results are identical. Therefore, it could be assumed, that lower levels of PAPP-A in the first trimester of pregnancy may be associated with an increased risk of GDM. In accordance with this, several studies have also shown that women with pre-existing diabetes mellitus (DM) have significantly lower PAPP-A levels than those without DM [54,55].

Nevertheless, few other investigations have not found such a relationship between PAPP-A and GDM [55,56,57,58,59]. This discrepancy is probably due to the lack of identical criteria and characteristics of the analyzed patients. For example, the study conducted by Husslein et al. has failed to demonstrate alterations in PAPP-A levels, but the authors investigated only women who developed GDM and needed insulin treatment at 11–14 weeks of gestation [58]. Hence, PAPP-A is not yet considered an early indicator of carbohydrate disorders during pregnancy and further studies are warranted to prove its predictive value.

## 4. PlGF

Placental growth factor (PlGF), first isolated in 1991, is a homodimeric glycoprotein belonging to the vascular endothelial growth factor (VEGF) family [60,61]. It is expressed mainly in the syncytiotrophoblast and cytotrophoblast of the placenta, but low levels have also been registered in the endothelial cells and bone marrow erythroblasts [62,63]. Since the placenta is the major source of PlGF, its circulating levels are markedly elevated during pregnancy, but the expression of PlGF alters at different stages of placental development [61]. PlGF has been shown to induce the proliferation, migration, and activation of endothelial cells, thus playing a central role in development and maturation of the placental vascular system and circulation [63].

In an uncomplicated pregnancy, PlGF concentrations in the first trimester are low, increasing from 11–12 weeks of gestation with a peak at the 30th week, after which there is a gradual decrease in its levels [64]. PlGF is considered as one of the major predictors of the occurrence of a complication, such as preeclampsia and fetal growth restriction [62,65].

Noteworthily, elevated levels of PlGF have been also observed in pregnant women with GDM. The possible explanation might be due to the fact that hyperglycemia affects angiogenesis and maternal hyperglycemia stimulates placental neovascularization [66]. Moreover, PlGF has been shown to play a role in development and the function of the placental vascular network, and there has been evidence that the placentas of women with GDM and pre-existing type 1 and 2 diabetes mellitus are characterized by a higher total area of the terminal villi of the placenta and an increased number of small vessels [67,68,69].

The changes in PlGF levels have been shown to be positively associated primarily with fasting blood glucose. This may also explain its predictive role in women who do not have risk factors for diabetes or pre-pregnancy IR [70,71,72,73]. Table 2 summarizes some of the studies showing a positive association between increased serum PlGF levels and GDM.

Although the cited data suggest a significant relationship between the serum PlGF concentrations and GDM, there are studies that have not demonstrated differences in PlGF levels between women who developed GDM and controls [74,75]. Tsiakkas et al. observed that maternal serum PlGF levels were even reduced in women with pre-existing diabetes [75]. Therefore, PlGF, like PAPP-A, is not yet considered as an early screening marker for the risk of GDM development.

## 5. Conclusions

GDM is a health problem associated with the risk of many complications for both mother and fetus, childbirth, and beyond. Hence, the early diagnosis and timely treatment of this condition is of pivotal importance. Nonetheless, the diagnosis of GDM is very often delayed as a result of the universal screening for carbohydrate disorders during pregnancy, carried out in the interval 24–28 weeks of gestation. The establishment of maternal-fetal medicine as an independent specialty has brought indisputable positive changes in the early diagnosis of many anomalies and diseases. Markers of placentation, such as PlGF and PAPP-A, are routinely tested in almost all pregnant women at the end of the first trimester of pregnancy as biomarkers of pre-eclampsia and aneuploidies. Therefore, it would be useful to evaluate their predictive potential also for GDM. The current review summarized the available data for the use of these markers as predictors of carbohydrate disorders in pregnant women. Despite a few conflicting results, most analyses have shown that low levels of PAPP-A and elevated levels of PlGF are significantly associated with a higher risk of GDM. However, further large-scale studies are needed to conclude whether these markers could be used in early screening for GDM.

## Figures and Tables

**Figure 1 medicina-59-00398-f001:**
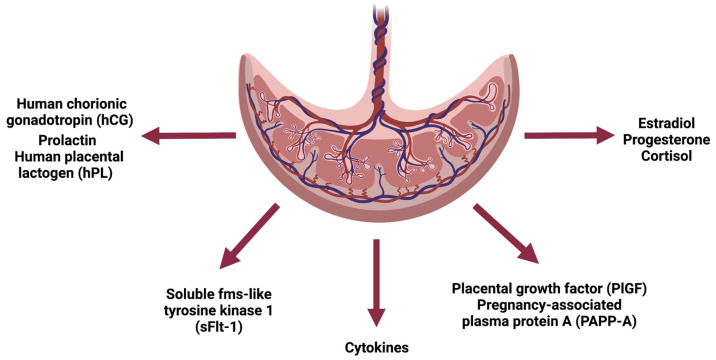
Schematic representation of the endocrine function of the placenta. Created with BioRender.com (accessed on 17 December 2022).

**Figure 2 medicina-59-00398-f002:**
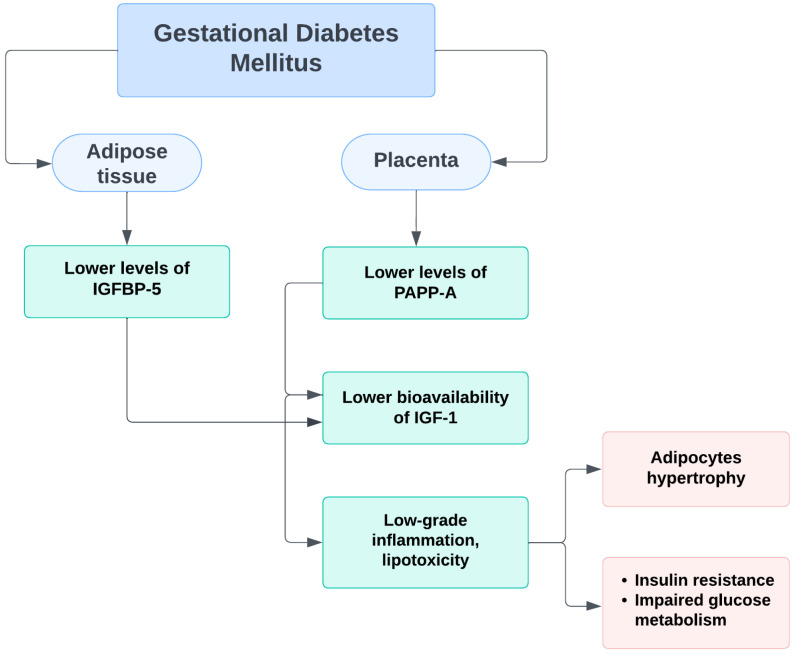
Schematic representation of the probable mechanism of PAPP-A impact on insulin resistance and glucose homeostasis during pregnancy.

**Table 1 medicina-59-00398-t001:** Studies showing an inverse relationship between serum PAPP-A levels and GDM risk.

Authors and Reference Number	Country	Year of Publication	Sample Size (Cases/Controls)	Study Design	Biochemical Markers		GDM	
					Tested Marker(s)	Time of Testing (Gestational Weeks)	Diagnosis	Time of Diagnosis (Gestational Weeks)
Ong et al. [39]	UK	2000	5584 (49/4297)	Cohort	PAPP-A; hGT	10–14	75 g OGTT	24–28
Benevetti et al. [40]	Italy	2011	459 (228/228)	Case-control	PAPP-A; hGT	10–13	50 g and 100 g OGTT	24–28
Lovati et al. [41]	Italy	2013	672 (307/366)	Case-control	PAPP-A; hGT	10–13	75 g OGTT	24–28
Spencer et al. [42]	UK	2013	7429 (870/6559)	Cohort	PAPP-A; hGT	10–13	75 g OGTT	24–28
Kulaksizoglu et al. [43]	Turkey	2013	(60/60)	Case-control	PAPP-A	11–13	75 g OGTT	24–28
Benevetti et al. [44]	Italy	2014	347 (retro.112/retro. -112) pros-18/pros-105	Retrospective and prospective case-control	PAPP-A; hGT	10–13	100 g OGTT	24–28
Wells et al. [45]	Australia	2015	1664 (274/1664)	Cohort	PAPP-A	10–14	75 g OGTT or 50 g GCT	After first antenatal visit (15 w) or at 26–28 w
Syngelaki et al. [46]	UK	2015	31 225 (787/30,438)	Case-control	PAPP-A; PlGF	11–13	75 g OGTT	24–28
Ferraz et al. [47]	Portugal	2016	2058 (205/1853)	Retrospective cohort	PAPP-A	11–14	75 g OGTT	24–28
Petry et al. [28]	UK	2017	821	Prospective and longitudinal	PAPP-A bioactive IGF	15 (on average)	75 g OGTT	28
Xiao et al. [48]	China	2017	1585 (599/986)	Case-control	PAPP-A; hGT	10–14	75 g OGTT	24–28
Ramezani et al. [49]	Iran	2017	250 (172/78)	Cohort	PAPP-A	11–14	75 g OGTT	24–27
Ramezani et al. [50]	Iran	2020	284 (201/83)	Prospective	PAPP-A	11–14	75 g OGTT	24–28
Ren et al. [51]	China	2020	99	Cohort	PAPP-A	NA	75 g OGTT	24–28
Caliskan et al. [52]	Turkey	2020	278 (120/158)	Case-control	PAPP-A; hGT	11–13	50 g and 100 g OGTT	24–28
Yanachkova et al. [53]	Bulgaria	2022	662 (412/250)	Retrospective case-control	PAPP-A	10–13	75 g OGTT	9–12 or 24–28

Abbreviations GDM: gestational diabetes mellitus; OGTT: oral glucose tolerance test; PAPP-A: Pregnancy-Associated Plasma Protein-A; hGT: human chorionic gonadotropin; PlGF: placental growth factor; N/A: not available.

**Table 2 medicina-59-00398-t002:** Studies establishing a significant relationship between elevated serum PlGF levels and GDM.

Authors and Reference Number	Country	Year of Publication	Sample Size (Cases/Controls)	Study Design	Biochemical Markers		GDM	
					Tested Marker(s)	Time of Testing (Gestational Weeks)	Diagnosis	Time of Diagnosis (Gestational Weeks)
Ong et al. [70]	UK	2004	482 (82/400)	Cohort	PlGF	11–14	75 g OGTT	24–28
Eleftheriades et al. [71]	Greece	2014	134 (40/94)	Case-control	PlGF; PAPP-A; hGT	11–14	75 g OGTT	24–28
Gocrem et al. [72]	Turkey	2020	158 (76/82)	Cross-sectional	PlGF	N/A	50 g GCT or 100 g OGTT	24–28
Yanachkova et al. [73]	Bulgaria	2022	662 (412/250)	Retrospective, Case-control	PlGF	10–13	75 g OGTT	9–12 or 24–28

Abbreviations PlGF: placental growth factor; GDM: gestational diabetes mellitus; OGTT: oral glucose tolerance test; GCT: glucose challenge test; PAPP-A: Pregnancy-Associated Plasma Protein-A; hGT: human chorionic gonadotropin; N/A: not available.

## Data Availability

Not applicable.

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
