# Peer review of "Placental Growth Factor and Pregnancy-Associated Plasma Protein-A as Potential Early Predictors of Gestational Diabetes Mellitus"

_medicina, 2023, doi:10.3390/medicina59020398_

Round 1

Reviewer 1 Report

The manuscript by Yanachkova et al is a great work and I am glad for the opportunity offered me to revise it.
It gives us a clear view about two biomarkers that can early predict GDM, but I have few comments:

Title: It would be better if you add “mellitus” at the end of the title

Typing:

               If you use an abreviation, please maintain the acronym (e.g., GDM (L38, L132-133) for Diabetes Mellitus (L44 etc), IR (L63) for insulin resistance (L66, L78), IGF (L112) for insulin-like growth factor (L125, L137 etc). Please review the whole manuscript and eliminate these discrepancies.

               Please explain the abreviation first time when you use them in text – PIGF and sflt-1 (L100)

               intrauterine fetal retardation” (L122) - did you mean intrauterine growth restriction?

Figure 2 – I suggest to try to make the schematic representation clearer by adding on the “addipose tissue” column the connection of low IGFBP-5 with low levels of circulating IGFs.

Table 1 – first of all, I think that the table should be connected to the information from the manuscript more smoothly. Secondly, I think that it lacks a method of writing, because you give no information about the criteria used for including the studies; so, I recommend to add the method or to rewrite the table after a systematic revision of the literature. The table, also, lack information about the number of patients, from the included studies, that had a modified OGTT at the evaluation period or, even better, from the 24-28 weeks TTGO.

L187 and L191 - at what type of diabetes do you refer – preexisting type 1 or 2 of diabetes mellitus (DM) or already diagnosed gestational DM?

Table 2 – I will discuss, again, about the method of inclusion for the studies from the table, because, without a systematic revision of the literature, there is hard to give a clear overview about the knowledge about the link between PlGF and GDM; so, I recommend to add the method or to rewrite the table after a systematic revision of the literature.

Moreover, starting from L196-198, it would be a more comprehensive review article, if it would include, also, the table with the studies that proves the lack of connection between PlGF and GDM.

Overall, my recommendation is to update the tables after a systematic revision of the literature.

Author Response

Dear Reviewer,

Thank you for the valuable comments and criticism.

We carefully followed your recommendations, implemented them, and corrected the manuscript accordingly. All changes that have been made are highlighted in the main text of the manuscript.

  • We added “mellitus” at the end of the title.
  • We have carefully corrected recommended expressions and abbreviations throughout the text.
  • We have added the recommended descriptive and supplementary sentences.
  • We added the connection between low IGFBP-5 with low levels of circulating IGFs in 2.
  • We have made a smooth transition to Table 1 in the text explaining the meaning of the table itself.
  • Our material is a narrative review. The role of placental markers for aneuploidy screening in early pregnancy and as predictors of complications such as preeclampsia and intrauterine fetal retraction has been actively considered by obstetricians. Our main idea for writing the present review is to present placental markers as biochemical indicators, to clarify their role in carbohydrate metabolism from a functional point of view, and, on this basis, to show their importance as predictors specifically of carbohydrate disorders.
  • The tables we present are only intended to illustrate and summarize the information in the main text. We wanted to present a short literature review connected to scientific activity confirming the relationship between these markers and carbohydrate disorders. That is also the reason why they are maximally simplified. Of course, following your recommendations, we added more studies proving our thesis.
  • We also present authors who do not confirm this thesis, but according to our main idea, we decided just to mention them in the main text.

Reviewer 2 Report

The manuscript summarizes current knowledge regarding two factors produced by placenta as a possible predictors of gestational diabetes. The manuscript is well-written and interesting; however, it can be noticeably improved.

One relevant paper that should be considered is not included in the Table 1 (PMID: 31335241). Also number of women with GDM and controls should be shown in the table.

My suggestion is that meta-analysis of all studies should be performed. Search strategy should be then described including key words used and this information might be presented as a flow chart.

Line 20 – delete critically

Line 28 – “On one hand”

Line 37 – delete generally, glucose instead of carbohydrate

40 – delete continue to

41-42 – reference should be provided

44 – abbreviation GDM is has already been introduced so you can use it, reference 2 should be replaced by more recent??

44-45 – please comment remaining 15 % of hyperglycaemia cases in pregnancy

49 – 24-28th week of gestation, please change accordingly elsewhere in the text

52 – delete “for the manifestation of carbohydrate abnormalities”

53-53 - “birth of a baby weighing more than 4000 g” should be replaced by “previous macrosomia”

54-55 – I suggest removing the sentence starting However, very often……

75 – change should be replaced by decrease

109 – replace identified by present

125 – abbreviation IGF can be used

132 – manifestation should be replaced by pathogenesis

186-187 – “women with diabetes” – do you mean GDM or different type(s)?

199 – what do you mean by pregestational diabetes?

200 – remove actively

Table 1 description – “Summary of the studies…”. Table 1 is very chaotic with present formatting. Landscape orientation on the separate page could help. Furthermore, Table 1 is not mentioned in the text.

Author Response

Dear Reviewer,

Thank you for the valuable comments and criticism.

We carefully followed your recommendations, implemented them, and corrected the manuscript accordingly. All changes that have been made are highlighted in the main text of the manuscript.

  • We have carefully corrected recommended expressions and abbreviations throughout the text.
  • We've added the recommended descriptive and supplemental sentences.
  • We have made a smooth transition to Table 1 in the text explaining the meaning of the table itself.
  • We've added the recommended article (PMID: 31335241 in Table 1.
  • Our material is a narrative review as stated in the aim. The role of placental markers for aneuploidy screening in early pregnancy and as predictors of complications such as preeclampsia, and intrauterine fetal retraction, has been actively considered by obstetricians. Our main idea for writing the present review is to present placental markers as biochemical indicators, to clarify their role in carbohydrate metabolism from a functional point of view, and, on this basis, to show their importance as predictors specifically of carbohydrate disorders. Following your advice, in the future, a comprehensive meta-analysis could be done.

Round 2

Reviewer 1 Report

It can be published after proofreading process.